

# Don't worry, be active: how to facilitate the detection of errors in immersive virtual environments

Sara Rigutti[1], Marta Stragà[1], Marco Jez[2], Giulio Baldassi[1], Andrea Carnaghi[1], Piero Miceu[2] and Carlo Fantoni[1]

[1] Department of Life Sciences, Psychology Unit "Gaetano Kanizsa", University of Trieste, Trieste, Italy
[2] Area Science Park, Arsenal S.r.L, Trieste, Italy

Corresponding authors
Sara Rigutti, srigutti@units.it
Carlo Fantoni, cfantoni@units.it

## ABSTRACT

The current research aims to study the link between type of vision experienced in a collaborative immersive virtual environment (active vs. multiple passive), the type of error one looks for during a cooperative multi-user exploration of a design project (affordance vs. perceptual violations), and the type of setting in which multi-user perform (field in Experiment 1 vs. laboratory in Experiment 2). The relevance of this link is backed by the lack of conclusive evidence on an active vs. passive vision advantage in cooperative search tasks within software based on immersive virtual reality (IVR). Using a yoking paradigm based on the mixed usage of simultaneous active and multiple passive viewings, we found that the likelihood of error detection in a complex 3D environment was characterized by an active vs. multi-passive viewing advantage depending on: (1) the degree of knowledge dependence of the type of error the passive/active observers were looking for (*low* for perceptual violations, vs. *high* for affordance violations), as the advantage tended to manifest itself irrespectively from the setting for affordance, but not for perceptual violations; and (2) the degree of social desirability possibly induced by the setting in which the task was performed, as the advantage occurred irrespectively from the type of error in the laboratory (Experiment 2) but not in the field (Experiment 1) setting. Results are relevant to future development of cooperative software based on IVR used for supporting the design review. A multi-user design review experience in which designers, engineers and end-users all cooperate actively within the IVR wearing their own head mounted display, seems more suitable for the detection of relevant errors than standard systems characterized by a mixed usage of active and passive viewing.

## INTRODUCTION

To achieve an efficient visualization of the external world functional to the detection of relevant 3D environmental features, the brain has to integrate retinal information with extraretinal and proprioceptive information about the observer's ego-motion

(*Wallach, 1987*; *Braunstein & Tittle, 1988*; *Ono & Steinbach, 1990*; *Wexler, 2003*; *Fetsch et al., 2007*; *Fantoni, Caudek & Domini, 2010*). Only when the changes in retinal projections are accurately accounted for the sensed ego-motion information, a stable perception of the environment can be established, and environmental features can be efficiently detected. To what extent the brain integrates retinal and extra-retinal information, and whether such integration provides performance advantages in a task which is typical of cooperative multi-user exploration of complex design projects aided by Immersive Virtual Reality (IVR) is still under debate (*Jaekl, Jenkin & Harris, 2005*; *Liu, Ward & Markall, 2007*; *Teramoto & Riecke, 2010*; *Chrastil & Warren, 2012*; *Fantoni, Caudek & Domini, 2014*; *Bülthoff, Mohler & Thornton, 2018*). In order to provide an empirical answer to this question, we rely on IVR, and compare active vs. passive seeking performance for different types of design errors. In two experiments we contrasted participants' performance resulting from the dynamic stereoscopic/cyclopean view of a complex 3D environment, during either an active exploration (i.e. active viewing condition) or a passive replay of the exact same optic information, which was self-generated by the active observer (i.e. passive viewing condition).

## Passive and active observers in standard collaborative IVR systems

IVR offers a new human–computer interaction paradigm, in which users actively participate in, and interact with a computer-generated environment. The immersive experience offered by technologies, such as cave automatic virtual environment systems or head mounted displays (HMDs), leads to a sense of presence in the virtual environment that is experienced to some extent as it were real (*Bowman & McMahan, 2007*). IVR opens up the possibility to reproduce a variety of contexts that in the real world would be risky, costly or unreachable. For this reason, IVR has been applied to a variety of fields, beyond the entertainment industry, like training, simulation, rehabilitation, tele-operation and others (*Durlach & Mavor, 1995*; *Brooks, 1999*; *Schell & Shochet, 2001*; *Tarr & Warren, 2002*). Traditionally, IVR systems have been developed for single users. The user wears the HMD, interacts within immersive virtual environment and self-generates a 3D dynamic optic information by controlling the viewpoint motion. Viewpoint motion can be obtained by a combination of ego-motion and control devices generally guided by hand (i.e. SpaceMouse, Oculus Touch, and Leap Motion). The single user approach has been relevant to support and improve the immersive experience in different contexts, such as design, aesthetics, and 3D objects' visualization (*Bayon, Griffiths & Wilson, 2006*; *Chen, 2014*). However, it disregards the possible co-presence of more users that share the same immersive virtual environment, as in the case of cooperative working activities. The co-presence of the users in immersive virtual environments is typical of design review activities, in architecture, engineering and shipbuilding. In these working environments, a multi-user IVR system is required to support the activity of design review participants, which is mainly based on the collaborative recovery of design errors (*Fernández & Alonso, 2015*). Nowadays an optimal collaborative IVR system, in which all participants in the design review task wear

their own HMD, sharing a common 3D environment through tele-immersion (*Tarr & Warren, 2002*), is quite far from being used in real working environments for technological and economical drawbacks (*Chen et al., 2015*). To overcome this issue, standard collaborative IVR systems are based on a mixed usage of active and passive viewing and supported by HMD and projection screens, respectively (*Bayon, Griffiths & Wilson, 2006*). The visual immersive experience is generated by one observer who wears the HMD (i.e. the project leader of the design review session) and moves his point of view by body movements (encoded by HMD's translations and rotations), and manual control (encoded by different types of devices). His/her virtual experience is paralleled in real-time on a wide screen for collaborators. In this kind of collaborative IVR system, a single participant actively views the scene through self-controlling the viewpoint translation within the immersive virtual environment, while all remaining participants passively observe the scene from an external viewpoint (*Bayon, Griffiths & Wilson, 2006*; *Shao, Robotham & Hon, 2012*). This mixed usage of simultaneous active and multiple passive viewing constitutes the standard system for remote collaboration nowadays, in which users from different geographical locations share virtual environments (*Jones, Naef & McLundie, 2006*; *Bassanino et al., 2010*; *Chen et al., 2015*; *Fleury et al., 2015*; *Tanaya et al., 2017*). Although this mixed setting provides a sustainable technological solution, it is characterized by a drawback. Indeed, there is now considerable evidence suggesting that active and passive observers, although receiving similar visual input, rely on a different visual experience (*Wallach, Stanton & Becker, 1974*; *Braunstein & Tittle, 1988*; *Wexler, Lamouret & Droulez, 2001*; *Fantoni, Caudek & Domini, 2010*; *Caudek, Fantoni & Domini, 2011*; *Fantoni, Caudek & Domini, 2012*; *Fantoni, Caudek & Domini, 2014*). This difference is known to be due to the dissimilar sensori-motor and cognitive information resulting from an active exploration of the spatial layout relative to the passive observation, see supplemental text in Text S1 for further details (*Sherrington, 1906*; *Ono & Steinbach, 1990*; *Wilson et al., 1997*; *Chance et al., 1998*; *Christou & Bülthoff, 1999*; *Harman, Humphrey & Goodale, 1999*; *Wang & Simons, 1999*; *Ujike & Ono, 2001*; *James et al., 2002*; *Peh et al., 2002*; *Von Helmholtz, 2002*; *Wilson & Péruch, 2002*; *Wexler, 2003*; *Jaekl, Jenkin & Harris, 2005*; *Naji & Freeman, 2004*; *Waller, Loomis & Haun, 2004*; *Ono & Ujike, 2005*; *Wexler & Van Boxtel, 2005*; *Colas et al., 2007*; *Liu, Ward & Markall, 2007*; *Waller & Greenauer, 2007*; *Fantoni, Caudek & Domini, 2010*; *Riecke et al., 2010*; *Teramoto & Riecke, 2010*; *Caudek, Fantoni & Domini, 2011*; *Meijer & Van Der Lubbe, 2011*; *Ruddle, Volkova & Bülthoff, 2011*; *Ruddle et al., 2011*; *Chrastil & Warren, 2012*; *Fantoni, Caudek & Domini, 2012*; *Fantoni, Caudek & Domini, 2014*; *Bülthoff, Mohler & Thornton, 2018*). On the basis of these claims, we formulated the main expectation at the basis of our study: the compatibility between retinal and extra-retinal information resulting from ego-motion, together with agency and intentionality, that are at the basis of active but not of passive viewing, which ultimately mimics a standard collaborative IVR environment, should enhance the allocation of attention to relevant features of the 3D spatial layout. We tested this expectation comparing a possible bolstering effect of active vs. passive viewing in collaborative IVR systems based on the mixed usage of simultaneous active and multiple passive viewing. So far, it remains

unclear whether this type of advantage occurs in task used in cooperative software based on IVR (*Bayon, Griffiths & Wilson, 2006*).

## GENERAL METHOD

We addressed our aim by using a task, which is typical in the design review of large scale digital mock-ups, like those employed in ship design (*Jones, Naef & McLundie, 2006*; *Fernández & Alonso, 2015*): namely a modified *hide and seek* task (*Chandler, Fritz & Hala, 1989*). In particular, independent groups, which comprised both an active and multiple passive participants, were asked to simultaneously seek for different types of design errors in two different views, namely a dynamic stereoscopic (for the active observer) and dynamic cyclopean (for the passive observers) of a 3D ship corridor. These views either resulted from an active exploration of the IVR (in active viewing condition) or from the passive replay of the exact same optic information self-generated by the active observer (in passive viewing condition). The active observer self-generated the 3D view combining head (through HMD rotations and translations) and hand (through SpaceMouse controller) movements from a sitting position which was similar to the one adopted by passive observers. As the observers performed the task in a multi-user modality, we encoded the number of detected errors at the end of the exploration/observation session (lasting 4 min). A series of screenshots (nine in Experiment 1 or 10 in Experiment 2, with only eight of them referring to actual design errors and the remaining serving as *catch* trials) were shown to the group of participants. After each screenshot, participants were asked to raise their hand to indicate whether the screenshot included a design error they saw during the exploration phase.

Importantly, our modified *hide and seek* task, beyond being applicable in both field (i.e. in the form of demonstrative group game during a science and technology exhibit as in Experiment 1), and laboratory settings (i.e. in the form of group experiment as in Experiment 2), involves relevant visual and cognitive components generally characterizing cooperative technological applications based on IVR. First, it involves visual spatial attention, which is required to perform a visual search of design errors within a complex IVR scene. Visual spatial attention with active exploration of the actively generated/passively observed visual layout regulates the detection of a target feature (i.e. the error) among all potential distracters. Error detection is often involved in cooperative multi-user activity as in the case of design review sessions in architecture, engineering and shipbuilding domains (*Fernández & Alonso, 2015*). Secondly, the detection of errors within a complex 3D environment implied that the observer must form a representation of the detected object (i.e. erroneous), retrieve from memory a conventional representation of the object and finally compare the two representations to decide whether the detected object is erroneous or not (*MacKay & James, 2009*).

As far as we know, no studies, to date have tested the effect of active vs. multi-passive viewing in such a type of task when different types of design errors are involved in the 3D layout. To this purpose, we followed the Norman's categorization of design errors in artifacts (*Norman, 1988*) and included two different types of design errors in our task

with different degree of dependence on knowledge about the environment structure (Fig. 1):

1. Affordance Errors in Fig. 1A (AE, high degree), based on affordance violation such as a door handle in the same side of the hinges;
2. Perceptual Errors in Fig. 1B (PE, low degree), based on violation of perceptual organization principles like good continuation (such as a misaligned handrail) and colour similarity (such as a blue handrail embedded within a layout of yellow handrail).

According to Norman (1988), the detection of AE is a knowledge-based process. Indeed, the detection of AE results from incongruence between the mental interpretations of artifacts, based on our past knowledge and the experience applied to our perception of an artifact. By contrast, the detection of PE is perceptual based, as it is knowledge independent and results from incongruence directly stemming from the level of perceptual attributes, which are automatically processed, thus not requiring any learning process (i.e. a colour or shape discontinuity in the 3D layout). An AE type of error can thus be detected if the observer possessed knowledge about the conventional structure of the target object (e.g. the observer must know how a door looks like and how it works in order to recognize that the door handle is misplaced). Otherwise, a PE type of error can be detected even in absence of knowledge about target object structure (e.g. the observer does not need to know how the handrail should look like in conventional setting in order to detect the error). Hence, we expected that the likelihood of detecting an error in our task would depend on the type of error (AE vs. PE) irrespectively from viewing conditions (active or passive). The likelihood of detecting AEs was expected to be smaller than the likelihood of detecting PEs (Expectation 1).

We selected the immersive virtual environment from a section of a digital mock-up of a ship, in order to keep our task representative for cooperative software based on IVR. In particular, we choose a 3D ship corridor suitable for real multi-user session of ship design review (Fig. 1D). The structure of the immersive virtual environment was chosen so that the exploration path of the active observers was sequential relative to the ordering of appearance of design errors. This structure assured us that during the hide and seek task different active observers would have: (1) followed similar pathways; (2) passed at least once through all of the relevant places needed for the potential detection of all errors.

In line with the procedures outlined by previous studies, we employed a new adapted version of a yoked design. In yoked design, the passive viewing is often obtained replaying the active exploration of the same scene generated by another participant, or by the same participant (Rogers & Rogers, 1992; Harman, Humphrey & Goodale, 1999; James et al., 2002). Differently from this conventional design, our adapted design mimicked the viewing conditions typical of collaborative IVR systems based on the mixed usage of simultaneous active and multiple passive viewings. Therefore in our design, the active exploration and the multiple passive observations took place simultaneously (Meijer & Van Der Lubbe, 2011). The group of passive participants observed on a large screen the dynamic layout generated by the active observer in real-time. This design

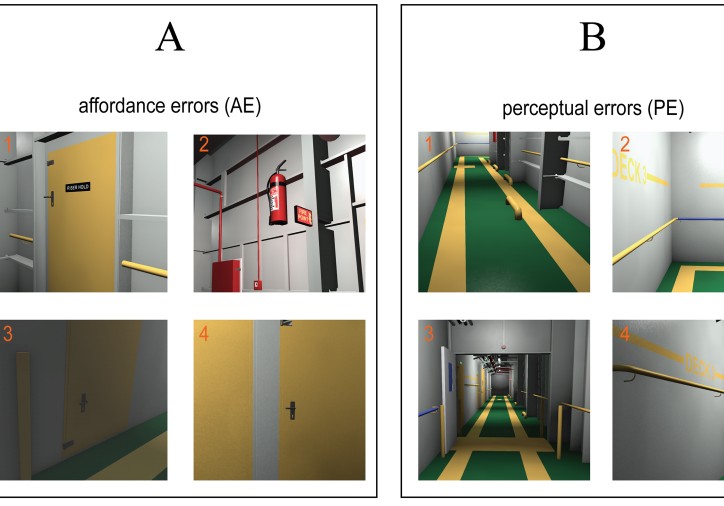

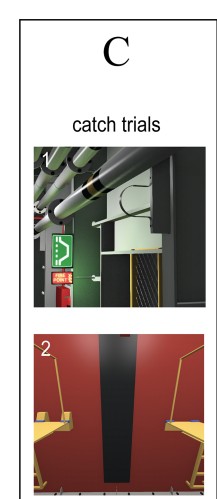

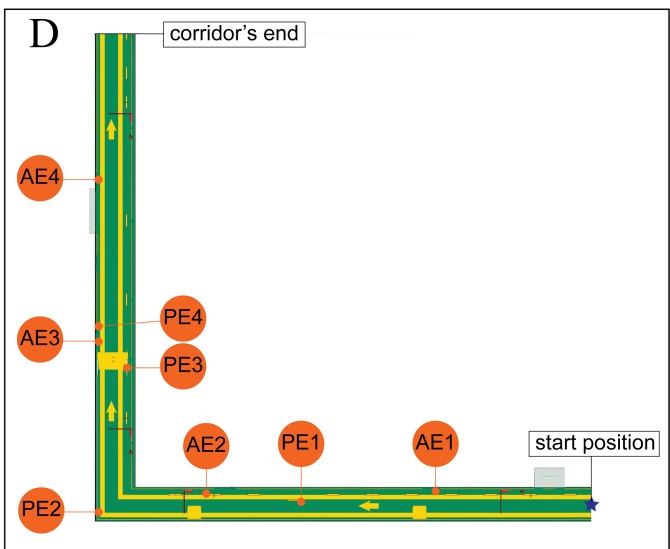

**Figure 1 Bird eye view of the immersive virtual environment and screenshots of design errors used in Experiment 1 and 2.** (A and B) screenshots of the eight actual design errors implemented in the digital mock-up of the corridor as presented to our participants during the response encoding phase, subdivided into the two types: affordance violations in (A) and perceptual violations in (B). Numbers indicate the relative ordering of appearance of violations along the immersive virtual environment explored during the task from the starting position to the corridor's end (D). (C) The two screenshots used as catch trials shown to the participants in an intermixed and randomized order together with the screenshots of the actual design errors used during the response-encoding phase of the experiments. Experiment 1 included the presentation of the only catch trial screenshot 1, Experiment 2 included the presentation of the both catch trials' screenshots (1 & 2). (D) A bird eye view of the immersive virtual environment, from the starting position (coded by the blue star) to the corridor's end (orange circles stand for design errors). The immersive virtual environment was the rendering of a digital mock-up of an L-shaped 3D ship corridor along which the eight design errors were sequentially implemented along the pathway the observer was required to travel (the numbering corresponds to their relative ordering of appearance along the pathway).

assures that, within each group, in the active and passive conditions the participants rely on the exact same visual input. This design overcomes the problem of the unbalanced visual exposure of different types of viewers (i.e. active and passive) which is typical

of yoked design, in which the participant passively views the optic array that results from the active exploration of a scene, which is generated by another (active) participant (*Chrastil & Warren, 2012*). Furthermore, differently from traditional yoked design, in which the passive view is replayed just after the recording of the active exploration (*Rogers & Rogers, 1992*; *Wexler, Lamouret & Droulez, 2001*; *Wexler, 2003*; *Fantoni, Caudek & Domini, 2010*), our simultaneous active and passive exposition to the dynamic 3D scene prevents us from potential threats to external and internal validity. Indeed, our simultaneous viewing involves a balanced (not unbalanced) temporal ordering of the viewing conditions.

Finally, we purposely unbalanced the amount of sources of information at which the active and passive viewers had access during our task, in order to assess the main aim of our study: to compare the effectiveness of different type of viewing resulting from collaborative IVR systems based on mixed usage of passive and active viewing. Our passive observers (subdivided in small groups) simultaneously viewed the same dynamic scene self-generated from a single active observer (Fig. 2), which was aided by stereoscopic vision (through HMD), extra-retinal and proprioceptive information regarding self-motion derived from head and hand movements, and cognitive control (see supplemental text in Text S1 for further details). We thus expected (Expectation 2) active viewing to be associated with a larger likelihood of design errors detection than passive viewing. This should result in an active vs. multi-passive error detection advantage. Notably, an opposite expectation occurs if the different amount of cognitive load involved in active exploration over passive observation is considered (Expectation 2b). In particular, the active exploration (relative to the passive observation), including demanding and distracting operation to intentionally move and translate the viewpoint, should result in a reduction of the attentional resources needed to perform the search task (*Liu, Ward & Markall, 2007*; see third paragraph in Text S1 for further details).

We validate the generalizability of the expected active vs. multi-passive advantage by varying the type of setting amongst Experiments, with Experiment 1 involving a field vs. Experiment 2 a laboratory setting. This variation had the purpose of delineating the limits within which the active vs. multi-passive advantage should fall. In particular, field settings (with a reduced control on extraneous variables) are known to induce social desirability biases more than laboratory settings as an instance of the reactivity to the experimental situation (*Crowne & Marlowe, 1964*; *Fisher, 1993*; *Shadish, Cook & Campbell, 2002*). Such reactivity in our task might lead to a ceiling effect given that the most socially desirable response was the one hitting not missing the error.

Both Experiments were approved by the Research Ethics Committee of the University of Trieste (approval number 84) in compliance with national legislation, the Ethical Code of the Italian Association of Psychology, and the Code of Ethical Principles for Medical Research Involving Human Subjects of the World Medical Association (Declaration of Helsinki). All participants provided their oral informed consent. The request of oral consent, formulated by the experimenter, made explicit that people not willing to participate in the session should simply not participate or not respond, without any consequence, and emphasis was put on the anonymous treatment of data which
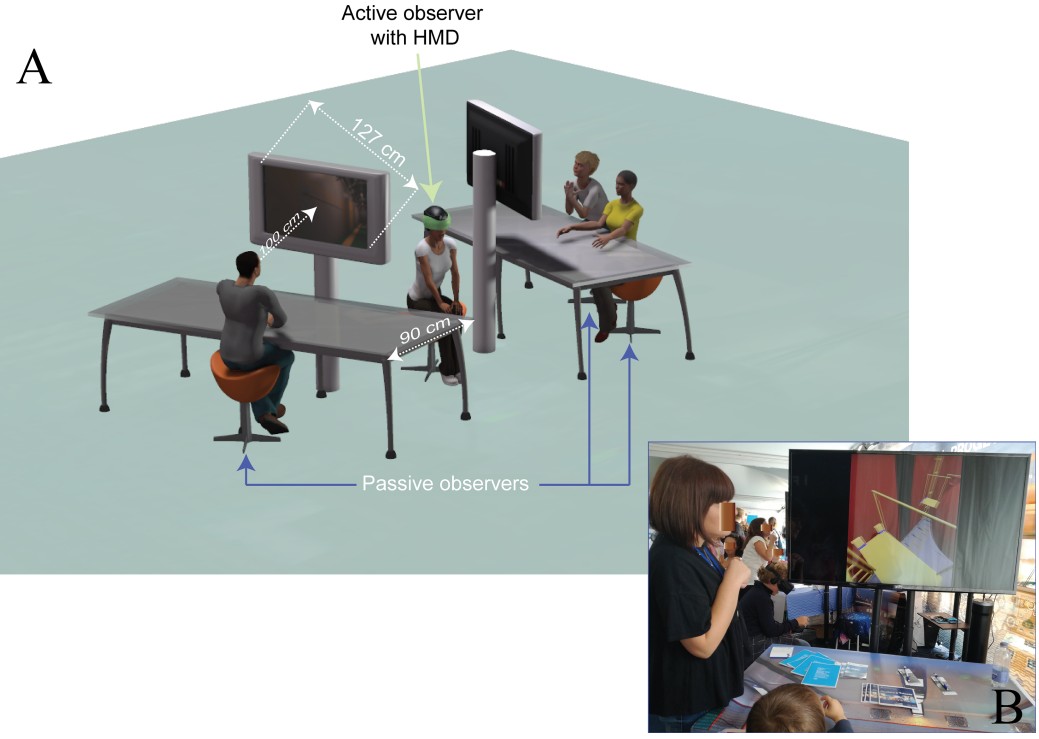

**Figure 2 Experimental setting of Experiment 1.** (A) Schematic view of the experimental setting with three observers (one active, green arrow, and two passives, blue arrows) implemented in the field setting during the science and technology exhibit of Experiment 1, with superposed the minimal distance the passive observers stand (as constrained by the 90 cm desk size), and their average viewing distance (100 cm when sitting with their elbows on the margin of the desk). This distance is taken from the two large LCD screens (127 cm diagonal), displaying in real time the exact same 3D, though monoscopic, view the active observer self-generated combining head movements and SpaceMouse control. In this example, the view produced by the active observer that the three passive observers are looking at in real time is consistent with PE1. (B) A photograph of the setting during the training session with the digital mock-up (a ship thruster) used to familiarize the active observer with the SpaceMouse and the passive observers with the 3D graphic. Notice that the laboratory setting of Experiment 2 reproduced in smaller scale the one shown in the current scheme, including smaller screens though smaller viewing distances in order to equate the two Experiments for the size of the passive displays in term of retinal sizes ($57.9° \times 34.5°$). Photo credit: Carlo Fantoni.

was part of group instructions at the beginning of session. Written consent (implying identification of every respondent) was redundant, given that the Experiments were carried out with visitors' groups scheduled according to the programme of the science exhibit within the field setting in Experiment 1 and with classes of psychology courses within the laboratory setting in Experiment 2. All participants present during the data collection sessions accepted to respond. The Ethics Committee of the University of Trieste thus approved the informed consent. Dataset is available as (Data S1).

## Experiment 1: hide and seek in a field setting

### Participants

Experimental sessions were conducted in a field setting during a science and technology exhibit at Trieste Next 2017 of a custom made IVR system used to support ship design

review. Visitors and scholars served as participants according to a well-defined scheduling of the science and technology exhibit lasting two days (self-selected sampling). All had normal or corrected to normal vision and were naive as to the purpose of the experiment. None of them reported specific visual diseases relevant for our task, like ocular dominance, colour blindness, and severe forms of eyestrain. A total of 98 participants took part in the study (mean age = 14.42, SD = 2.76, range [12, 20]), subdivided in 10 small groups of variable sizes ranging from a minimum number of five to a maximum number of 12. The groups were recruited one after the other during the exhibit. Six groups were composed by middle-school students ($n = 61$), two groups by high-school students ($n = 22$) and two by undergraduate students ($n = 15$). A total of 10 active participants vs. 88 passive participants were thus recruited, being each group composed by only one active participant. The active observer was selected among those participants who were willing to wear HMD, reported some previous experience with 3D gaming and low levels of motion sickness. This was assessed by orally asking them about previous experience of motion sickness symptoms according to *Kennedy et al. (1993)*: oculomotor (eyestrain, difficulty focusing, blurred vision, and headache), disorientation (dizziness, vertigo), and nausea (stomach awareness, increased salivation, and burping).

### Apparatus and stimuli

The active participants' head motions were tracked in real time using an Oculus Rift CV1 HMD (with a FOV of $93° \times 104°$ and a frame rate of 90 fps) connected with a PC equipped with an Intel Core i7 7700K 4.20 GHz processor with 16 GB RAM and nVIDIA GeForce GTX 1080 Ti graphics card. The PC controlled the simultaneous active and passive visualization of our complex immersive virtual environment and sampled the tracker. The immersive virtual environment was rendered through an IVR system designed by Arsenal S.r.L and was updated in real time according to a combination of head and hand movements of the active observer (*Jez et al., 2018*). In particular, the translation/rotation of the point of view through the immersive virtual environment was supported by the combination of head movements and manual control through a 3D Connexion SpaceMouse Pro, with four Degrees of Freedom (DOF; i.e. transition in the directions of three directional axes—forwards/backwards, left/right, up/down—and rotation on the vertical axis). Accelerometers within the HMD together with two camera motion sensors converging with an angle of about 40° on the active observer's position at rest (at a distance of about 1.5 m) were used to calculate the x, y, z coordinates of the observers viewpoint. These coordinates defined the head movements of the active observers and were used by our programme to update in real time the geometrical projection of the 3D graphic. A precise visualization of the virtual environment was achieved by carefully calibrating the 3D layout to the height and the inter-ocular distance of each active observer following the Oculus Rift procedure. The virtual reality experience generated by the HMD was paralleled on two large LCD screens (Samsung 50″ J5200 Full HD LED set at a screen resolution of $1,024 \times 768$ pixels) connected with the PC. The two LCDs simultaneously displayed a cyclopean view of the exact same dynamic optic information generated by the movements of the active observer within the IVR.

Figure 1 provides a bird eye view of the virtual environment (Fig. 1D) used for the *hide and seek* task together with the design errors implemented throughout the path (screenshots, in Figs. 1A and 1B, for AEs and PEs, respectively). Our virtual environment consisted of a custom-made digital mock-up of an L-shaped 3D ship corridor. The corridor was 2.8 m high, 2.4 m large, and was composed by two straight segments: a first 35 m long segment connecting the starting position to the corridor's curve (Fig. 1D, horizontal part of the corridor), and a second 40 m segment connecting the corridor's curve to the corridor's end (Fig. 1D, vertical part of the corridor). The corridor included different features, like: floor texture (a green floor with lateral yellow stripes and horizontal arrows indicating the directions), long pipes on the ceiling (with different sizes and colours), doors (five along the wall and one louvered door in the middle of the corridor), security systems (such as fire extinguisher, warning lights), signals (e.g. 'fire point', exit), handrails, bollards, and several technical equipment. The screenshots of the eight design errors actually included in the digital mock-up of the corridor (as presented to the participants to the experiment during the response encoding phase) are shown in Figs. 1A and 1B, together with the screenshot used as a catch trial (Fig. 1C). The catch trial was a misplaced pipe holder that was not implemented as an actual error in the digital mock-up of the corridor though being visually consistent with the features of the corridor. Within our experimental design, valid screenshots (i.e. those depicting actual errors) were equally subdivided into the two types: AE and PE (Figs. 1A and 1B, respectively).

Affordance errors included (Fig. 1A) a door handle placed too high and in the same side of the hinges (AE1), a fire extinguisher placed too high (AE2), a door handle placed too low and in the same side of the hinges (AE3), and a door handle in the same side of the hinges (AE4). PE included (Fig. 1B) a misaligned bollard (PE1), blue handrail embedded within yellow handrails (PE2), a louvered door missing an half (PE3), and a misaligned handrail (PE4). Three independent judges categorized the eight errors into the two categories, with an inter-rater agreement of 94%. By combining our two types of valid screenshots (AE/PE) with the two viewing conditions (active/passive) we obtained four experimental conditions of a mixed factorial design.

As shown in Fig. 1D, the design errors depicted within the screenshots were distributed all along the length of the 3D ship corridor, at a distance from the starting point of 16 m (AE1), 25 m (PE1), 31.5 m (AE2), 40 m (PE2), 49.5 m (PE3), 52 m (AE3), 53 m (PE4), 63 m (AE4).

The spatial arrangement of the setting is schematized in Fig. 2A, with Fig. 2B showing a photograph of the actual exposition context taken during a training session on a digital mock-up different from the one used during the experiment. The active observer comfortably sat on a stool right in between the two LCD screens, whereas the passive observers were arranged in front of the LCD screens comfortably sat on stools at a viewing distance of about 100 cm (Fig. 2A). A 90 cm wide desk was interposed in between the passive groups of participants and the screens in order to control at the best the viewing distance (observers were required to sit on the stools posing the elbows on the margin of the desk). At the viewing distance of 100 cm the retinal size subtended by the 3D

dynamic layout when presented in full screen was of about 57.9° horizontally and 34.5° vertically.

## Procedure

The experimental procedure included six phases lasting overall 10 min.

First, participants were informed about the experimental setting and the instructions of the modified *hide and seek* task using a Power Point presentation. The task was described as a *game*, in which the best-performing participant would be awarded with a gadget. This procedure was also aimed to motivate each single participant to perform at his/her best. To this purpose, an experimenter first explained to the entire group of participants the role they were about to endorse in the game. Participants were then introduced with their spatial disposition depending on the group they belonged: passive participants (one in front of the LCD to the left of the active observer, the other in front of the LCD to the right of the active observer, as by Fig. 2A) or active participant (right in between the two groups of passive participants, Fig. 2A). They were then informed about the aim of the game: 'to search, find and memorize for design errors they would have encountered during the exploration of an immersive virtual environment'. Participants were also informed that at the end of the exploration they would have been administered a brief serial presentation of screenshots. After each screenshot's presentation they would have been asked to raise their hand if the screenshot reported an error they saw during the exploration phase of the game.

Second, participants were then assigned to an active ($n = 10$) or passive ($n = 88$) role.

In the third phase, the group of participants was familiarized with the immersive virtual environment they were going to view during the game. During this familiarization phase, all participants watched on the LCD screen a 1 min video clip showing the same 3D ship corridor they would have been exposed during the exploration phase of the game but without including design errors. Participants were instructed to carefully watch the video clip. In order to encourage subjects to be as accurate as possible in seeking design errors, the experimenter stressed that in the video clip design errors were absent. This information also provided the subjects with a hint on the design errors they would have successively encountered during the experimental phase.

Fourth, the active observer was briefly trained in the usage of the SpaceMouse to control the translation of his/her point of view. The training was performed keeping static the HMD and visualizing its view on the LCD screen. A digital mock-up different from the one used during the experimental phase was used for such a phase (see an example in Fig. 2B). The observer following the instruction of the experimenter was trained for about 1 min on the main commands of the SpaceMouse (rotation, lateral and back/forth viewpoint translation). During this phase, in order to similarly train the passive observers with the 3D complex structure of the environment, the passive observers were required to look at the same scene the active observer was looking at.

Fifth, the experimental phase took place lasting about 5 min (1 min of instruction + 4 min of game). Active and passive observers were first required to keep their positions as by instruction (Fig. 2). The experimenter then focused the observers' attention on the

experimental task and instructed them to memorize the errors they found throughout the exploration phase of the game without disclosing their identification. Participants were thus explicitly discouraged to use neither verbal (i.e. claims) nor non-verbal behaviour (i.e. gestures) as soon as they detected a design error. The exploration phase of the game then started: the active observer was required to begin moving freely along the 3D ship corridor (now including the eight design errors) in search for design violations. This phase was terminated after 4 min of exploration. On average, this duration led to travel along the corridor back and forth for about one time.

Finally, the sixth response-encoding phase took place. Participants were serially presented with nine screenshots: the eight design errors presented serially in the order they appeared along the pathway from the start position to the corridor's end plus the catch trial presented in a random serial position. As by initial instructions, participants were then required to raise their hands after each screenshot if the screenshot reported an error that they found during the 4 min exploration phase of the game: the experimenter registered the number of raised hands as well as the participants who raised their hand. At last, participants were thanked, debriefed and awarded with a gadget. Importantly, participants, throughout the game, were never aware of the number and types of design errors included in the 3D ship corridor.

### Results

We analysed the individual likelihood of detecting a design error as an index of performance in our modified *hide and seek* task following *Knoblauch & Maloney (2012)* by applying a generalized linear-mixed effect model (*glmm*) with a probit link function to the whole set of individual binary responses on valid screenshots (1 = error detected; 0 = error not detected). In particular, we used a *glmm* with the type of design error (AE vs. PE) and the viewing condition (Active vs. Passive) as fixed factors, with by subject and by error (i.e. the eight valid screenshots) random intercepts and by subject random slope for type of design error (*Baayen, Davidson & Bates, 2008*). Importantly, this type of analysis has been proved to be optimal for a research in which the distribution of responses and participants across conditions is inevitably unbalanced as in the case of our design in which the proportion of individual responses resulting from the active observers was of about 0.105. Two-tailed *p*-values were obtained from type 3 *F*-statistics with the denominator's DOF calculated by subtracting from the total number of observations the model's DOF minus one. As indices of effect size we reported the partial eta-square ($\eta_p^2$), and the concordance correlation coefficient, *rc*. Following *Vonesh, Chinchilli & Pu (1996*; but see also *Rigutti, Fantoni & Gerbino, 2015*) such a latter index provides a reliable measure, in the −1 to 1 range, of the degree of agreement between observed values and values predicted by generalized linear-mixed effect models. Finally, we reported Cohen's *d* as a standardized measure of significant difference between means.

Disregarding catch trials, the data analysis was based on the 100% of active participants' individual responses to valid trials (*n* = 80 resulting from the combination of 10 active observers and eight errors), and 97% of passive participants' responses (*n* = 680 out

of the total of 704 responses, resulting from 85—three excluded from the total of 88—passive observers and eight errors).

We removed three passive observers from the analysis because of the application of two exclusion criteria both aimed at minimizing any possible biasing effect of social desirability intrinsic in our field setting:

1. relative the sample mean deviation, one passive participant achieved an individual error-detection proportion that deviates from the one of the active observer of the group he/she belonged more than $\pm$ 2.5 SD;
2. two passive participants provided a positive response (i.e. raised their hand) after the catch trial presentation.

According to the result of the a priori power analysis, the total sample size of 760 observations, resulting from the application of the exclusion criteria and entered onto the *glmm* model in order to estimate the global active vs. multi-passive advantage, resulted to be large enough to rely on reliable statistical conclusions. We performed the power analysis following *Faul et al. (2009*; see the section 5.1) and using the generalized linear regression tool of G* Power (3.1.9.2 version) for testing logistic regression models based on the $z$-score distributions for the proportion as those implied in our *glmm* model (with the probit as a link function). In particular, the required sample size resulting from the analysis was equal to 695, which was indeed smaller than our total sample size of 760 observations obtained from the total number of valid binary responses of our 95 participants in our repeated measures design (with eight trials—one valid screenshot each—for each participant). The required sample size was obtained as follows. Based on a pre-experimental planning of the size of our group sessions we estimated an average proportion of active participants within each group of about 0.11 (eight passive observers for every single active observer). We entered this ratio as the best estimate of the expected value of our binomial distribution of the binary encoding of the type of viewing condition variable (1 = active; 0 = passive). We assumed an odds ratio of about 1.86, resulting from an expected gain in the detection of error proportion of the active over the passive condition of about 0.15 with $p$(H0) = 0.5, a power of 0.80, and no correlation between our two predictors (i.e. the type of viewing and the type of error). Notably, the required sample size resulted to be still smaller (728) than our total sample size of observations when calculated entering into the analysis (as the average of the binomial distribution associated to our binary set of type of viewings) the actual ratio (0.105) that we observed from the number of active observations (80) and the total number of observations in our design (760).

The pattern of average error detection proportion as a function of the type of design error ($x$-axis) and viewing condition (colour coding as by the legend) shown in Fig. 3A is only in part consistent with our expectations. Expectation 2 (not 2b) indeed holds for AE but not for PE.

This was confirmed by the results of the *glmm* analysis ($rc$ = 0.52, 95% CI [0.48, 0.55]) revealing an unexpected trend towards significance of the interaction between type of

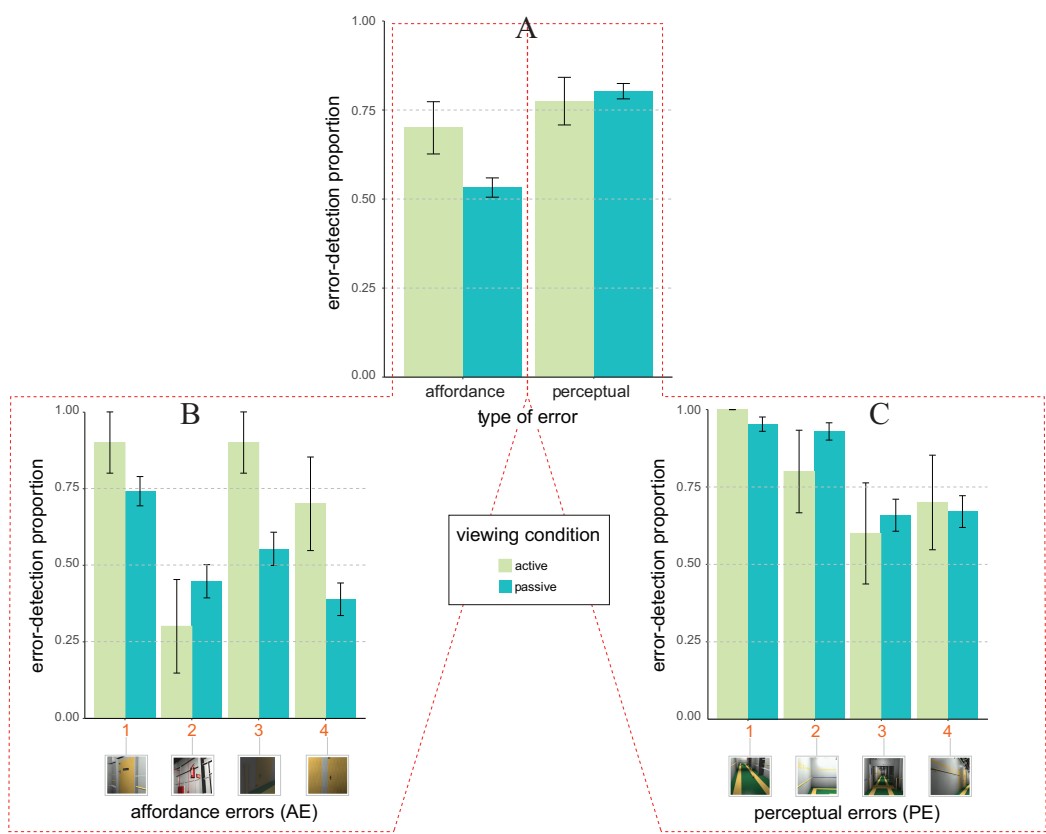

**Figure 3 Error detection proportion in Experiment 1 (field setting).** (A) Mean and SE values of error detection proportion in active (green) vs. passive (blue) viewing conditions as a function of the type of error (*affordance* vs. *perceptual*) on the abscissa. (B and C) Mean and SE values of error detection proportion in active (green) vs. passive (blue) viewing for the four types of affordance errors (B) and the four types of perceptual errors (C). The numbers along the abscissa indicate the relative ordering of error's appearance along the exploration path of the 3D ship corridor, from the starting position to the end (same encoding of Fig. 1). 

design error and viewing condition ($F_{1,\,751} = 3.90$, $p = 0.048$, $\eta_p^2 = 0.005$, 95% CI [0.000, 0.017]). Such a trend was due to the fact that the likelihood of detecting AEs was larger for active ($M = 0.70 \pm 0.07$, with $M$ indicating average proportion of error detection plus/minus one standard error), rather than passive ($M = 0.53 \pm 0.02$) observers (consistent with Expectation 2, not with Expectation 2b). This difference produced a marginally significant *glmm* estimated active vs. multi-passive error detection advantage of about ($0.22 \pm 0.23$, $z = 1.61$, one-tailed $p = 0.053$, $d = 0.34$). Differently, the likelihood of detecting PEs was similar for active ($M = 0.77 \pm 0.07$) vs. passive ($M = 0.80 \pm 0.02$) conditions thus producing a non-significant active vs. multi-passive advantage ($-0.03 \pm 0.15$; $z = 0.43$, one-tailed $p = 0.332$).

The *glmm* analysis further revealed that Expectation 1 was only in part validated by our pattern of data as revealing a main effect of the type of error ($F_{1,\,751} = 5.04$, $p = 0.025$, $\eta_p^2 = 0.007$, 95% CI [0.000, 0.020]) with an overall *glmm* estimated facilitation for PE over AE of about $0.31 \pm 0.17$. However, this facilitation was moderated by the type of viewing, as qualified by the type of viewing × type of error interaction. The facilitation

for PE over AE was indeed present only for passive, but not for active viewing conditions. For the passive viewing condition the likelihood of detecting an error was indeed smaller for AEs ($M = 0.53 \pm 0.02$), rather than PEs ($M = 0.80 \pm 0.02$) type of errors ($z = -2.38$, one-tailed $p = 0.009$, $d = 0.60$). No such a difference was found for the active viewing condition in which the likelihood of detecting an error was instead similar for AEs ($M = 0.70 \pm 0.07$) and PEs ($M = 0.77 \pm 0.07$) type of errors ($z = 0.47$, one-tailed $p = 0.318$).

A further post hoc *glmm* analysis was carried out on the distribution of the likelihood of errors detection amongst our two types of errors (Figs. 3B and 3C for affordance and perceptual, respectively). This analysis revealed the active vs. multi-passive advantage for AE type of error was mainly due to two out of the four AEs: AE3 and AE4 (Fig. 3B). We indeed found a significant active vs. multi-passive error detection advantage for AE3 ($z = -2.10$, one-tailed $p = 0.018$, $d = 0.71$) and AE4 ($z = -1.84$, one-tailed $p = 0.033$, $d = 0.64$), but not for AE1 ($z = -1.18$, one-tailed $p = 0.120$) and AE2 ($z = 0.92$, one-tailed $p = 0.178$). No reliable differences were instead found for any one of the four PEs (PE1: $z = -0.07$, one-tailed $p = 0.470$; PE2: $z = 1.24$, one-tailed $p = 0.107$; PE3: $z = 0.37$, one-tailed $p = 0.354$; PE4: $z = -0.21$, one-tailed $p = 0.415$; Fig. 3C).

### Discussion

In the current Experiment, we found a trend towards an active vs. multi-passive viewing advantage with AEs but not PEs. As we hypothesized, the failure to find a difference between active and multi-passive observers in PEs can be the result of an overall ceiling effect. According to Expectation 1, such a ceiling was more likely to occur on PEs rather than AEs, given that PEs was expected to be easier to be detected than AEs. Such a response ceiling is likely to be induced by the type of field setting used in the current experiment favouring positive (hand raised) rather than negative (hand not raised) responses. Positive responses were indeed more socially desirable than negative responses in our task, thus producing a social desirability bias. Given these mixed results, the field experimental setting and the heterogeneous sample that we employed in Experiment 1, we ran a second experiment using a more controlled setting possibly reducing social desirability bias.

## Experiment 2: hide and seek in a laboratory setting

*Would results be similar (as those of Experiment 1) when responses at the basis of error detection are less biased by social factors?* In order to answer such a question, we performed Experiment 2 that was a replication of Experiment 1 in a VR laboratory (not a field setting), with a sample of participants more uniform in term of age than the one used in Experiment 1, and with smaller groups of active/passive observers. Furthermore, relative to Experiment 1, in Experiment 2 we further reduced the possible social desirability bias, introducing:

1. a method to validate the truthfulness of the errors reported by the active observers signalled by his/her screenshots;
2. an additional catch trial, now reaching a 25% of catches over the total number of valid trials ($n = 8$);

3. the removal of the ludic dimension as a context within which the participants were asked to perform the task (i.e. the task was not described as a game and participants were not awarded with gadgets for their performance).

Importantly, the new catch trial was intentionally extracted from an immersive virtual environment apparently different from the one tested during the exploration session. Such an apparent difference should implicitly inform the participants that the series of screenshots presented during the response-encoding phase did not necessarily include actual design errors. This should further reduce the tendency to respond according to social desirability.

### Participants

A total of 100 undergraduates of the University of Trieste (mean age = 20.60, SD = 3.29, [18, 40] range, 77% female), all with normal or corrected to normal vision, participated in the experiment in return for course credits. According to our pre-experimental questioning (same as in Experiment 1) our participants did not suffer of any specific visual diseases relevant for our task (ocular dominance, colour blindness, and severe forms of eyestrain). Participants were subdivided in 17 groups with variable size, each of which ranged from three to eight. Groups were tested in individual sessions and were formed following the same procedure used in Experiment 1. A total of 14 groups were composed by first year bachelor psychology students ($n = 91$), and three groups each composed by three participants included second year master psychology students ($n = 9$). Participants were assigned to active ($n = 17$) and passive ($n = 83$) conditions following the same selection criteria used in Experiment 1.

### Apparatus and stimuli

The experiment was conducted in a VR laboratory equipped with the same experimental apparatus of Experiment 1, with three major differences:

1. The stereo projection system was driven by an MSI laptop instead of a PC, equipped with an Intel Core i7 7820HK 2.90 GHz processor with 32 GB RAM and nVIDIA GeForce GTX 1070 graphics card;
2. the LCD screens used to simultaneously display the dynamic cyclopean view of the 3D ship corridor explored by the active observer, were smaller (22″ not 50″, Samsung S22E450M LED set at a screen resolution of 1,024 × 768);
3. the overall viewing distance of the passive observers was settled at about 43 cm (not 100 cm) so to equate the displays in term of retinal size.

In order to provide a method that corroborates the reliability of the errors reported by the active observers, the IVR system was aided by a functionality that allows to store in real time the screenshots of the cyclopean views of the active observer. A screenshot was stored during the exploration as soon as the active participant pressed the R button of the SpaceMouse.

The spatial disposition of participants within the setting, and the immersive virtual environment explored during the task (the 3D L-shaped ship corridor, Fig. 1D) together with the design errors (Figs. 1A and 1B) were the same as in Experiment 1. The screenshots used during the response-encoding phase were the same as in Experiment 1 (8 valid screenshots, 4 AEs + 4 PEs, plus one catch trial screenshot), with one additional screenshot used as catch trial. This additional screenshot displayed a missing holder in a ship thruster (Fig. 1C, screenshot 2). This catch trial screenshot was apparently different from all of the other screenshots for colour properties, being reddish not greenish. Such a difference in colour purposely magnifies the un-relatedness of some trials with the 3D ship corridor thus reducing the tendency to favour positive over negative responses in our task. In Experiment 2, the same 2 × 2 mixed factorial design of Experiment 1 applied, with four experimental conditions resulting by the full factorial combination of two types of design errors × 2 viewing conditions.

### Procedure

The procedure was similar to the one used in Experiment 1 with the major difference that the ludic dimension was now intentionally removed from the task so to reduce the social desirability bias. The procedure included the same six phases described in subsection 2.1.3 and lasted almost the same time (10 min). Major differences regarded:

1. the first phase, in which task was described as a game, was omitted from the current instruction;

2. in the second phase, in which, although following the same selection criteria used in Experiment 1, albeit a relatively larger number of participants were assigned to the active ($n = 17$) and the passive ($n = 83$) conditions;

3. the fifth experimental phase, that although requiring the participants to perform the same modified *hide and seek* task of Experiment 1 (lasting 4 min), it included an additional task for the active participant (to push the R button of the SpaceMouse as soon as he/she detects a design error during the exploration phase);

4. the sixth response encoding phase that was conducted following the exact same procedure as in Experiment 1, but that included the serial presentation of 10 (not 9) screenshots.

At last, we collected demographic information (age and gender), and participants were thanked and debriefed following the same procedure used in Experiment 1.

The debriefing in Experiment 2 also served the purpose of gathering information from those active observers that during the exploration phase did take a number of screenshots that differs from those signalled by the raising of their hand. A total of 10 observers out of the total of 17 took a screenshot of all of the errors they reported during the response-encoding phase. The remaining seven observers either took no screenshot at all ($n = 1$), or took one ($n = 5$) to three ($n = 1$) screenshot less than the number of errors they reported during the response encoding phase. The average number of correspondence between screenshots and errors reported by the raising of hands was of about 82%.

This showed that our response method was reliable. In the debriefing phase, all seven participants with a smaller correspondence between the screenshots they took and the errors they reported claimed that they might have: (1) forgotten about pushing the SpaceMouse button during the exploration phase; (2) pushed the SpaceMouse button accidentally or to signal relevant parts of the environment in addition to mere errors. In line with this latest claim, the analysis of the screenshots reveals that observers in general used the SpaceMouse button not only to signal a detection error but also to signal a salient part of the 3D environment. The average individual proportion of screenshots relative to the total number of actual errors ($n = 8$) was indeed reliably larger than the average individual error detection proportion ($0.72 \pm 0.09$ vs. $0.48 \pm 0.05$, $t = 3.91$, d$f = 16$, $p = 0.001$, $d = 1.95$). Consequently, we decided not to use the screenshots as an additional exclusion criterion for our participants, given that our result suggested that they were used for a mixture of purposes (accidental button press, to signal a detection error, or to signal a relevant part of the environment).

### Results

As in Experiment 1, we analysed the individual likelihood of detecting a design error as an index of performance in our modified *hide and seek* task by applying the same *glmm* model (with *probit* as a link function) to the whole set of individual binary responses on valid screenshots (1 = error detected; 0 = error not detected), with type of design error (AE vs. PE) and viewing condition (Active vs. Passive) as fixed factors, and with by subject and by error random intercepts and by subject random slope for type of design error. We used the same indices of effect size as in Experiment 1 in order to support the reliability of our statistical effects. Disregarding catch trials, the data analysis was now based on the 100% of active participant's responses to valid trials ($n = 136$ resulting from the combination of 17 active observers and eight errors), and on the 96% of passive participant's responses ($n = 640$ out of the total of 664 responses, resulting from 80—three excluded from the total of 83—passive observers and eight errors). After the application of the same exclusion criteria used in Experiment 1, we removed from the analysis three passive observers who raised their hand when one of the two catch trials was presented.

Performing the same a priori power analysis of Experiment 1 (assuming the same power, and odds ratio), our total sample size of 776 observations entered onto the *glmm* model resulted to be large enough to rely on reliable statistical conclusions. The sample size was indeed larger than the one used in Experiment 1, and more balanced across viewing conditions, with the planned proportion of active participants within each group raised to 0.2, which resulted from planned groups that on average were formed by one active participant out of four passive participants. The increased balance over the two viewing conditions delivered a required sample size of 427, that was much smaller than the observed one. Notably, the required sample size was still smaller (473) than the one we used also entering into the analysis (as the average of the binomial distribution associated to our binary set of type of viewings) the actual ratio (0.175) that we observed from the number of active observations (136) and the total number of observations in our design.

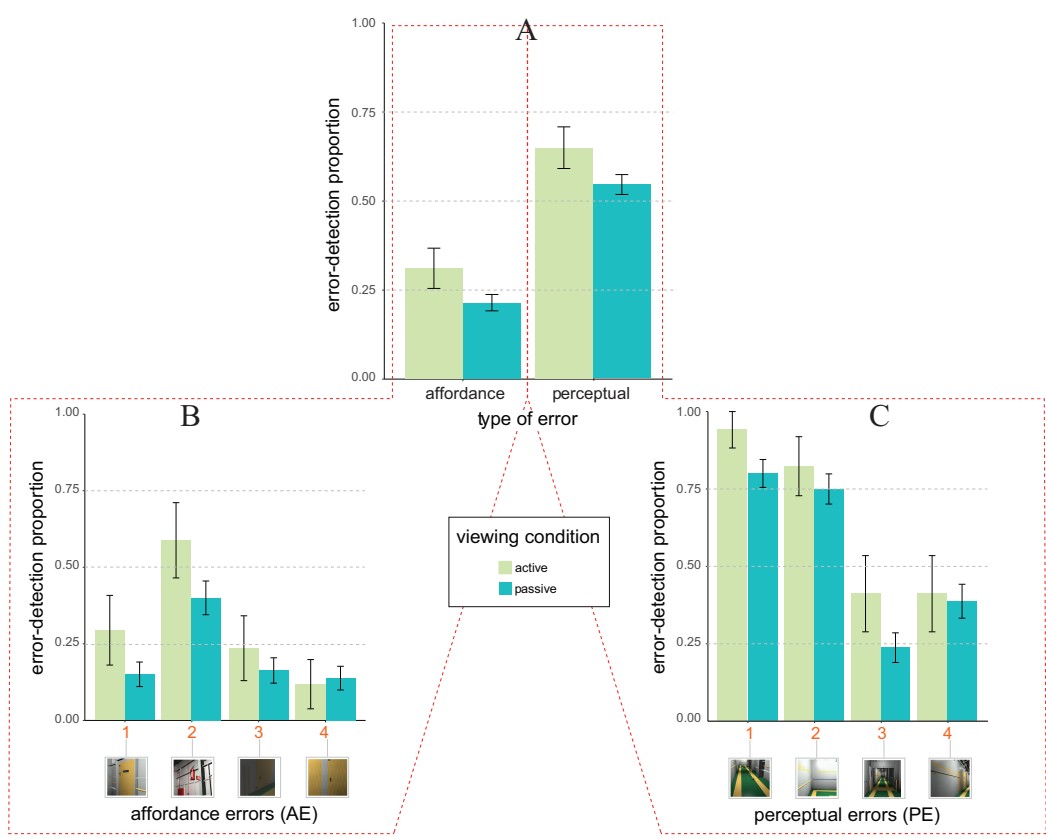

**Figure 4 Error detection proportion in Experiment 2 (laboratory setting).** (A) Mean and SE values of error detection proportion in active (green) vs. passive (blue) viewing conditions as a function of the type of error (*affordance* vs. *perceptual*) on the abscissa. (B and C) Mean and SE values of error detection proportion in active (green) vs. passive (blue) viewing for the four types of affordance errors (B) and the four types of perceptual errors (C). The numbers along the abscissa indicate the relative ordering of error's appearance along the exploration path of the 3D ship corridor, from the start position to the end (same encoding of Fig. 1).

Figure 4A depicts the pattern of average error detection proportion as a function of the type of design error (*x*-axis) and viewing condition (colour coding as by the legend). The pattern is apparently different from the one of Fig. 3A (Experiment 1), being now consistent with both Expectation 1 (main effect of the type of design error), and Expectation 2 (main effect of viewing condition). Furthermore, differently from Experiment 1 the type of design error × viewing condition is now absent.

These observations were confirmed by the statistical *glmm* analyses ($rc = 0.58$, 95% CI [0.54, 0.61]). Indeed, as consistent with Expectation 2 (not 2b), *glmm* results showed a significant main effect of the viewing condition ($F_{1, 767} = 5.02$, $p = 0.025$, $\eta_p^2 = 0.007$, 95% CI [0.000, 0.019]), with a larger likelihood of detecting an error in active ($M = 0.48 \pm 0.04$), rather than in passive viewing condition ($M = 0.38 \pm 0.02$). This effect was quantified by an active vs. multi-passive error detection advantage of about $0.15 \pm 0.13$ ($z = 2.15$, one-tailed $p = 0.016$, $d = 0.20$). Expectation 1 was supported by the significant main effect of type of design error ($F_{1, 767} = 8.23$, $p = 0.004$, $\eta_p^2 = 0.011$, 95% CI [0.002, 0.026]), with a larger likelihood of detecting an error for PE ($M = 0.56 \pm 0.03$), rather than for

AE ($M = 0.23 \pm 0.02$). Unlike Experiment 1, the interaction between type of design error and viewing condition was not statistically significant ($F_{1,\,767} = 0.12$, $p = 0.727$, $\eta_p^2 = 0.000$, 95% CI [0.000, 0.005]).

As in Experiment 1 we further performed a post hoc *glmm* analyses on the way in which the likelihood of errors detection was distributed amongst our two types of errors (Figs. 4B and 4C for affordance and perceptual, respectively). This analysis revealed that the origin of the active vs. multi-passive advantage was likely to be qualified by two out of four AEs (Fig. 4B, AE1, $z = -1.52$, one-tailed $p = 0.064$, $d = 0.38$; and AE2, $z = -1,43$, one-tailed $p = 0.077$, $d = 0.38$), and two out of four PEs (Fig. 4C, PE1, $z = -1.45$, one-tailed $p = 0.073$, $d = 0.37$; and PE3, $z = -1.39$, one-tailed $p = 0.083$, $d = 0.39$), all of which showed a trend towards significance concerning the active vs. passive error detection advantage.

### Discussion

Experiment 2 revealed that both the active vs. multi-passive viewing advantage predicted by Expectation 2 (not 2b), and the effect of facilitation due to the type of error predicted by Expectation 1, did occur under a laboratory setting. Joining data from both Experiments revealed an overall reduction of the tendency to provide a positive response in Experiment 2 relative to Experiment 1. The likelihood of detecting an error strongly decreased in the laboratory setting of Experiment 2 ($M = 0.40 \pm 0.02$), relative to the field setting of Experiment 1 ($M = 0.67 \pm 0.02$). Such a difference was confirmed by the significant main effect of the Experiment ($F_{1,\,1523} = 92.53$, $p < 0.001$, $\eta_p^2 = 0.057$, 95% CI [0.040, 0.077]), when it was included as an additional fixed factor in the *glmm* analysis. The direction of this effect might be accounted for by social desirability, which more likely biases responses towards positive detection in field rather than in laboratory settings.

## CONCLUSIONS

We reported two experiments on the link between the type of vision one might experience in a collaborative virtual environment (active vs. multiple passive), the type of error one might look for during a cooperative multi-user exploration of a complex design project (affordance vs. perceptual violations), and the type of setting—manipulated through Experiments—within which a multi-user activity is performed (field in Experiment 1 vs. laboratory in Experiment 2). Our two experiments demonstrated that the likelihood of error detection within a complex 3D immersive virtual environment is characterized by an active vs. multi-passive viewing advantage (consistent with our Expectation 2). In particular, we found that such an advantage depends on multiple sources of information, like:

1. The degree of knowledge dependence of the type of error the passive/active users were looking for (*low* for perceptual vs. *high* for affordance violations), as the advantage tended to manifest itself irrespectively from the setting for affordance, but not for perceptual violations. This was suggested by the main effect of the type of viewing in Experiment 2 vs. the type of viewing × type of error interaction in Experiment 1.

This difference was characterized by an anisotropy: a facilitation in the detection of PEs over AEs (consistent with our Expectation 1) which is particularly evident under the laboratory setting (Experiment 2), given that it also emerges in the active vision conditions. We interpreted such facilitation as a by-product of the relative complexity of the encoding process supporting the detection of the error. AEs would involve a more complex encoding process than PEs to be detected, as their encoding depends on previous knowledge about the structure of the object, while PEs being based on mere violation of perceptual organization principles as good continuation and colour similarity does not requires the access to knowledge (*Norman, 1988*; *MacKay & James, 2009*). Furthermore, the overall facilitation of PEs over AEs is consistent with a strand of evidence suggesting that in tasks involving objects' recognition, artifact recognition is slowed down given that they automatically activate multiple levels of information, from manipulative to functional (*Gerlach, 2009*; *Anelli, Nicoletti & Borghi, 2010*; *Costantini et al., 2011*; *Fantoni et al., 2016*).

2. The degree of social desirability possibly induced by the setting in which the task was performed, as the active vs. multi-passive advantage occurred irrespectively from the type of error in the laboratory (Experiment 2) but not in the field (Experiment 1) setting. This anisotropy was qualified by an overall enhancement of the likelihood of reporting an error detection inducing a response ceiling in the field rather than in the laboratory setting. This was consistent with the fact that social desirability biases occur more often in field than laboratory settings (*Crowne & Marlowe, 1964*; *Fisher, 1993*; *Shadish, Cook & Campbell, 2002*).

Taken together, these results can be reconciled with the somewhat mixed findings stemming from literature on active/passive exploration/observation for error detection and way finding (*Liu, Ward & Markall, 2007*; *Chrastil & Warren, 2012*; *Bülthoff, Mohler & Thornton, 2018*). The peculiar information to which active but not passive viewers have access during exploration/observation like stereopsis (monoscopic cyclopean view in passive viewers), extra-retinal and proprioceptive information from ego-motion (absent in passive viewers), and agency/intentionality (absent in passive viewers), would have contributed to determinate a superiority of active vs. passive vision in the detection of errors in our complex 3D environment. However, this superiority might result both from the correspondence between retinal and extraretinal egomotion signals, and from the correspondence between retinal and proprioceptive signals from hand movement used to control the viewpoint motion within the immersive virtual environment. This latest information component linked to manual control is indeed known to positively affect perceptual performance in both 3D (*Harman, Humphrey & Goodale, 1999*; *James et al., 2002*) and 2D space (*Ichikawa & Masakura, 2006*; *Scocchia et al., 2009*).

Notably, following *Liu, Ward & Markall (2007)*, a superiority of active over passive vision is only apparently in contrast with the lower cognitive load to which our passive, rather than active, observers were subjected to during our task (with only the active observer being involved during the error detection task also in complex activities required by the active exploration of the environment, like controlling/moving the point of view,

deciding where to look for etc.). It was indeed possible that in our task such an active vs. passive disadvantage was overshadowed by an active vs. passive advantage produced by the compatibility between visual and non-visual information, together with agency and intentionality, that characterize active (not passive) vision. These peculiar components might have enhanced the allocation of visual spatial attention towards relevant features of the complex 3D spatial layout we used in our Experiments. According to this interpretation, a perspective point for future studies would be to test whether this overshadowing could be minimized by reducing the complexity of the 3D environment. This reduction of complexity could lead to a reversed pattern of advantage in line with the one we found in the current study: an active vs. multi-passive disadvantage.

In general, our findings may make a useful contribution to the literature on error detection within 3D environments, technology, working sciences and methodology. Our results, indeed allowed us to provide a first tentative response to a relevant though still debated research question: *How the effectiveness of the interaction with a complex 3D environment is affected by the different types of devices (HMD vs. screen) mediating the immersive experience offered by VR technologies in collaborative contexts?*

In the present study we empirically answered to this question that poses a novel research problem, namely the *multi-user vision problem*. This problem involves the understanding of whether a self-generated stereoscopic and immersive view of a complex layout leads to a more effective representation of the 3D scene compared to the passive reply of the same optic information simultaneously displayed on flat screens to multiple passive observers. This problem is rooted into actual working collaborative contexts, such as design review sessions in working domains (e.g. architecture, engineering, and shipbuilding) in which the co-presence of more users, which share through cooperation the same project, typically occurs. These design review sessions nowadays are supported by standard collaborative IVR systems based on a mixed usage of passive and active viewing combining HMD and projection screens (*Bayon, Griffiths & Wilson, 2006*). In general, previous studies have not yet provided conclusive evidence on the superior advantage of a type of viewing over another during 3D interaction in collaborative immersive virtual environments. Active and passive viewing, although being based on the same visual input, have indeed access to substantially different sources of information (*Wexler, Lamouret & Droulez, 2001*; *Wexler, 2003*; *Fantoni, Caudek & Domini, 2010*; *Caudek, Fantoni & Domini, 2011*; *Fantoni, Caudek & Domini, 2012*). Importantly, here we approached for the first time the *multi-user vision problem* using a novel adaptation of the yoked paradigm (*Rogers & Rogers, 1992*; *Fantoni, Caudek & Domini, 2014*). Our adapted yoked paradigm reproduced the simultaneous active/multi-passive conditions occurring in standard collaborative IVR systems: observers subdivided into groups, each composed by one active and multiple passive observers, were indeed asked to simultaneously search for and find (a modified *hide and seek task*) design errors within a complex 3D ship layout. However, although our paradigm mimicked the viewing conditions typical of standard collaborative IVR systems, it was not fully representative of the real interaction conditions characterizing the working environment in which IVR systems are generally used. In our paradigm, indeed active and passive participants

were not allowed to interact and communicate each other during the modified *hide and seek task*. This limit leaves open the question of the generalizability of our active vs. multi-passive viewing advantage to the real working domain, that could be addressed by future research.

As a perspective point for the future development of cooperative software based on immersive virtual environments, we believe that our study might provide a relevant hint. A multi-user design review experience in which designers, engineers and end-users, all cooperate actively within the same IVR wearing their own HMD, seems more suitable for the detection of relevant errors, than standard systems characterized by a mixed usage of active and passive viewing. This point is rooted in the pioneering work of *Tarr & Warren (2002)*, proposing the development of tele-immersive systems for the support of the design process, in which users do share a common 3D graphical workplace. This idea is particularly relevant for the implementation of future technological solution based on tele-immersive systems for the support of design review that, according to *Fernández & Alonso's (2015)* claims, should minimize implementation errors within projects.

## ACKNOWLEDGEMENTS

We would like to thank all participants and Valentina Piccoli for helping with participants during NEXT 2017.

### Funding

This work was supported by a POR-FESR 2014-2020 (SIDRAN project, UE and Friuli Venezia Giulia) grant to Carlo Fantoni and Piero Miceu. The funders had no role in study design, data collection and analysis, decision to publish, or preparation of the manuscript.

### Grant Disclosures

The following grant information was disclosed by the authors:
POR-FESR 2014-2020 (SIDRAN project, UE and Friuli Venezia Giulia).

### Competing Interests

The authors declare that they have no competing interests.

### Author Contributions

- Sara Rigutti conceived and designed the experiments, performed the experiments, analysed the data, contributed reagents/materials/analysis tools, prepared figures and/or tables, authored or reviewed drafts of the paper, approved the final draft.
- Marta Stragà conceived and designed the experiments, performed the experiments, analysed the data, contributed reagents/materials/analysis tools, authored or reviewed drafts of the paper, approved the final draft.
- Marco Jez conceived and designed the experiments, performed the experiments, contributed reagents/materials/analysis tools, approved the final draft.

- Giulio Baldassi performed the experiments, analysed the data, approved the final draft.
- Andrea Carnaghi contributed reagents/materials/analysis tools, authored or reviewed drafts of the paper, approved the final draft.
- Piero Miceu conceived and designed the experiments, contributed reagents/materials/analysis tools, approved the final draft.
- Carlo Fantoni conceived and designed the experiments, performed the experiments, analysed the data, contributed reagents/materials/analysis tools, authored or reviewed drafts of the paper, approved the final draft.

## Human Ethics

The following information was supplied relating to ethical approvals (i.e. approving body and any reference numbers):

The experiments carried out in the study were approved by the Research Ethics Committee of the University of Trieste (approval number #84, date: 11/13/2017).

## Data Availability

The raw data are available in a Supplemental File.

## Supplemental Information

Supplemental information for this article can be found online at http://dx.doi.org/10.7717/peerj.5844#supplemental-information.

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
