# Peer review of "Don’t worry, be active: how to facilitate the detection of errors in immersive virtual environments"

_PeerJ, doi:10.7717/peerj.5844_

## Round 0.1 · original submission · Major Revisions

Based on the reviews you should complete major revisions. Please address all the comments of the reviewers.

Reviewer 1 ·

Basic reporting

This article is original, well written and easy to follow. I only suggest reviewing the points listed below in order to improve the text.

A. Please define the acronyms listed below:
1. Line 226-227: Please define the acronyms into brackets “(A vs P)”;
2. Line 497: Please define the acronyms “DE3” and “DE4”;

B. Please check out the reference to the figures in the text. Specifically at lines 211-213, Figure 1 is wrongly referenced. I believe Fig.1B and 1C might be Fig.1A and 1B instead.

C. A few typos to correct:
1. Line 47: Please change “Mavpor” with “Mavor”;
2. Line 109: Please change “Arns” with “Arms”;
3. Line 137: Please change “Bulthoff” with “Bülthoff”
4. Line 154: Please change “ideothetic” with “idiothetic”;
5. Line 263: Please change “active.” with “active”;
6. Line 418: Please change “exemplar” with “example”;
7. Line 531: Please change “Undergraduates'” with “Undergraduate”;
8. Line 548: Please change “reduced” with “reduction”;
9. Line 551: Please change “allow” with “allows”;
10. Line 722: Please change “thorough” with “through”.

D. I suggest improving the description of the participants taking part to the study. I have a few questions regarding this point. Did you make any preliminary study on the tested population? For example, did you check if the participants had comparable seek and hike errors recognition ability or did you test the participants for color sensitivity? Did you consider these factors in the depiction of the experimental setting?

Experimental design

no comment

Validity of the findings

no comment

Reviewer 2 ·

Basic reporting

This work cannot be published in its present form. Its appears that the paper has been translated from its original language into English. There are few spelling mistakes however the grammar is entirely unacceptable and the use of words in almost every paragraph make interpretation of the experiment reported extremely difficult to understand.

Experimental design

The 'Active' versus 'Passive' paradigm, as the authors report, has been used in numerous previous experiments. From most of these experiments it seems to me that active exploration produces better performance than passive and there are various reasons for this.
The experimental design is perhaps reasonable although the fact that the sessions occurred during visits by students etc. on a haphazard basics leads me to be believe that the conditions were not highly controlled.

Validity of the findings

There is little novelty here apart from the fact that a comparison is made between affordance and perceptual error in passive versus active viewing.
The data is reasonably well stated given the obvious problems with the use of the English language.
I cannot see that the conclusion adds anything to our general knowledge.

Annotated reviews are not available for download in order to protect the identity of reviewers who chose to remain anonymous.

·

Basic reporting

The article is written in English and is generally clear, although there are numerous instances where there are grammatical or stylistic errors that detract from the work. It is recommended to have the manuscript reviewed again in detail to catch these. Note that the results are clear so issues with language do not affect the ability to review the work.

The abstract and introduction could be more direct and focus with respect to the aim of the paper. While the authors provide a reasonably comprehensive review of relevant previous work, I found it difficult to keep in mind what the purpose of the experiment was.

Experimental design

The experimental design was clear. As noted above, the research question could be better defined prior to the experimental design.

Validity of the findings

The statistical approach appears sound, although as it is not a standard approach it is good to guide the reader through the approach used, which the authors mostly do. While impact and novelty are not of concern, and negative/inconclusive results are permitted, I will require all results to report measures of effect size and power analyses. This is both important to understand just how strong the reported effects are (and their likelihood for being reproducible) as well as to assess those effects that were not found to be significant. Reporting of effects sizes and statistical power are particularly important as there were some unexpected results found be the researchers.

Additional comments

I thought that this is a useful paper on what appears to be a novel research design. If the authors make more of an effort to streamline the paper and make it more approachable it will help readers tremendously to best understand the work. The addition of measures of effect size and power analyses will significantly strengthen the results section and temper the interpretation of the results accordingly.

---

## Round 0.2 · Minor Revisions

The MS is improved but still needs remaining minor revision.

Reviewer 1 ·

Basic reporting

the authors reviewed the paper and improved the English form of it. The manuscript is overall improved and more readable.

Experimental design

The authors improved the description of both their aim and of the method used. I really appreciated the details added about the participants' selection and characteristics.

Validity of the findings

The authors improved the results section through an improved statistical analysis.

Additional comments

The authors improved overall the manuscript that to my opinion should be accepted for publication.

·

Basic reporting

The language used in the manuscript is now clearer. I will leave this to the editor to decide if the quality of the language is appropriate for this journal.

Experimental design

While the vestibular system is certainly involved during self-motion here, it is not acting alone. Consequently the authors should temper their use of "vestibular" as there is no attempt to isolate the vestibular system from other systems that are active during self-motion compared to passive motion.

Validity of the findings

It is very helpful that the authors have included measures of effect size throughout the manuscript. I have remaining issues with the reporting of statistics and the interpretation of the results:

L444, L449: report the full p value (don't round). If the p value is actually 0.05 exactly then there is no effect, but perhaps a trend. Given that the sample size is well powered, but the effect size is very small, this is most likely not an effect and should be interpreted this way with caution

L621: if reporting that all of these are trends and below 0.09, the full range should be reported and the language should not be "marginally significant" but rather trended towards reaching significance. In small underpowered designs trends can be important, but in well powered designs these really should be reported as not being significant.

What I would like to see in the results, discussion and abstract then is more effort in being skeptical of the results. For those results that are clearly significant and with large effect sizes then treat them as such. For the remaining trends and very low effect sized results these really should be reported as providing no or very little evidence.

---

## Round 0.3 · Minor Revisions

I have been asked to take over from the previous editor, who is currently unavailable, to finalize the manuscript. It had previously been returned for 'minor revisions', and the reviewers and editor were generally happy with the last version of the manuscript. I have reviewed the history of reviews/comments and concur that the previous comments are mostly dealt with satisfactorily, and do not need to be sent out for another round of review.

I have also read the final ms carefully, and made annotated comments. These are mostly stylistic/language suggestions. There are some incorrect reference formatting, some spurious capitalizations, etc. Please consider these as suggestions that will help improve the readability of the manuscript.

In reading the manuscript, I have about about five more serious concerns that I would hope can be addressed in what I would hope will be the final revision, either in the text where the issues occur, or in the discussion. I think each of these are really fairly minor; insofar as each could be dealt with with minor editing of claims or an additional sentence or two of discussion. I hope the authors do not get discouraged by these because the ms has already been returned for minor revision, but these are concerns I likely would have raised on earlier versions if I had been the editor or a reviewer.

1. Claims of Naturalistic/ecological setting or task. I don't see how doing this at the NEXT 2017 conference constitutes an ecological setting. Furthermore, the fact that it was a realistic environment (instead of the sparse 3d spaces sometimes used) is nice, but it does not make it ecological or naturalistic. I'd argue that an ecologically-valid study would occur with teams working on the design of actual systems. To deal with this concern, please play down the claims that this is ecologically-valid, as I do not think the distinction between the two experiments is one of ecological validity.

2. Comparisons between experiments.The fact that the results of the test were not exactly replicated for perceptual problems could stem from many things, including random variation, instruction difference, participant difference, as well as the different distractions from the conference event. I did not see any statistical test that established whether the two results indeed differed, so it is hard to draw conclusions about the study context having an effect, although this was implied in trying to explain why the results differed. I'd suggest the most likely account is that because the active viewing N is so small and there was already large variability in the subject pool (including middle school students). Thus you have very low precision on that estimate for study 1; study 2 was a bit better, but still not great.

To deal with this, I would recommend being careful about the discussions comparing the two studies; focusing on the things that were replicated, and avoiding claims about differences in the result (because none were established). You might simply suggest that given the population of study 1, the null results are not unexpected even if there is a true difference. See also point 3 below.


3. Power analysis. Related to the above explanation, I think also that the power analysis may be misleading or maybe incorrect. For example, a power analysis for a two-group t-test with n=90 and 10, power =.8, (similar to study 1) indicates that an effect size (eta^2) of above .8 is required. So with one of the groups having low numbers, it would be unlikely to find even strong differences all the time:

> pwr.t2n.test(n1=90,n2=10,power=.8,alternative="less")

t test power calculation

n1 = 90
n2 = 10
d = -0.834608
sig.level = 0.05
power = 0.8
alternative = less

Thinking about it another way, in study 1, one could do a yoked t-test where the active viewer is compared to the average of the passive viewers in the same condition, to factor out the variable effect of group/session. I'm not sure if the mixed effect model is doing this, but it may be similar. Nevertheless, this would give you a one-sample N of about 10, which would require an even larger effect size of about 1.0

> pwr.t.test(n=10,power=.8,type = "one.sample")

One-sample t test power calculation

n = 10
d = 0.9960043
sig.level = 0.05
power = 0.8
alternative = two.sided

To reverse this, one can compute the power from the design and effect. Several of the effects had effect sizes of .6 to .8 maybe, so we can see, using a eta^2 of .75:

> pwr.t2n.test(n1=90,n2=10,d=.75)

t test power calculation

n1 = 90
n2 = 10
d = 0.75
sig.level = 0.05
power = 0.6056443
alternative = two.sided

So, the experiment 1 design has a power of about .6, so it is pretty likely that a design like in experiment 1 would produce a null effect even if a large effect existed.

I'm not sure how to deal with this. It is not clear how the power analyses were done as reported in the paper, and so if some other argument can be made using an alternative power analysis, that is fine. More importantly, it gives maybe the most reasonable explanation for why the results did not replicate across experiments, so one doesn't have to make arguments about the thing being called a game or try to make inferences about the context it was studied under (lab or expo).

4. The yoking problem. I read the discussion of the yoking issues carefully, and I don't dispute that the design is one reasonable way to test this. However, there remain demand characteristics, because the active viewer knows his/her friends are watching--the viewing is thus not only done for themselves. Thus, the looking patterns may differ from what would occur if they were doing this in isolation. Of course, this is akin to how it would work in the workplace, but I'd suspect that if this were used in the workplace, designers would not be prohibited from talking to one another either. Probably one person would be 'driving' and other people would be telling him or her things like 'look back at that doorway'. Given the mouse and occulus is fussy to use, the lead designer or senior customer--the one who is really in charge--may be a passive viewer.

Anyway, to deal with this, I would recommend discussing the generalizability of these experiments as a limitation, and/or suggestion for follow-up, as part of the discussion. For example, the active session could be done in isolation, and then played back to individual passive viewers; or an active-passive group could be tested with verbal interaction.

5. Claims about user-centered design. This is pretty minor, but the authors use 'user-centered design' in a way that is at odds with how it is typically used in the human factors design field. I believe the intent was to distinguish design for a single-user versus for a group. This should be stated more carefully. Moreover, the experiments really do not test anything about user-centered design, or single- vs. multi-user design. I.e., there really are no design manipulations that are tested. So this argument is ultimately distracting and can be downplayed or removed. See comments in annotated manuscript.


Summary.

I have examined the manuscript, and have attached an annotated version that includes suggestions of minor and more serious concerns. I have listed my five biggest concerns above. I think each can be handled with minor edits to the manuscript; I see no need to change major analyses or experiments.

---

## Round 0.4 · accepted · Accept

I reviewed the edits you made and think they reasonably respond to the feedback. I better understand how you did the power analysis.

I have one comment regarding the power/sampling issue. I think my point is still valid--you have a sampling issue with only 1/10 of your participants seeing the active condition. The main argument against this was that you included all measures of individuals, not just one. But that is not really the case--you are trying to generalize across the distribution of people who might sit in that seat, not the distribution of individual responses. But the use of subject-based intercepts in the GLMM is how you captured this repeated factor, and so you still had 1/10 of the intercepts associated with participants in the active control group. Repetition only helps you estimate the individual's mean with greater precision; to estimate the population mean better, you need to sample more people. Otherwise, I could get a good estimate of the average height of people on the planet by measuring my own height a million times. As far as I can tell, the GLMM is done appropriately, I can't be too sure about the power analysis, but that is only useful for forecasting the size of the experiment needed, and so it is sort of a moot point (other than providing an explanation for why you may not have found the result, I guess)

Thus, it is not surprising if you fail to replicate one aspect of the results across similar experiments, and the simplest account is one of statistical power--nothing else requires explaining.

Despite this comment, I think you have adequately dealt with this issue, although if you think this comment might lead you to make minor edits to the discussion of power analysis forecasting, please do so.

#